# Morphological and Biophysical Study of S100A9 Protein Fibrils by Atomic Force Microscopy Imaging and Nanomechanical Analysis

**DOI:** 10.3390/biom14091091

**Published:** 2024-08-31

**Authors:** Ana P. Carapeto, Carlos Marcuello, Patrícia F. N. Faísca, Mário S. Rodrigues

**Affiliations:** 1BioISI—Biosystems and Integrative Sciences Institute, Faculdade de Ciências, Universidade de Lisboa, Campo Grande, 1749-016 Lisbon, Portugal; carlos.marcuello@ehu.eus (C.M.); mmrodrigues@ciencias.ulisboa.pt (M.S.R.); 2Departamento de Física, Faculdade de Ciências, Universidade de Lisboa, Campo Grande, 1749-016 Lisbon, Portugal; 3Biofisika Institute (CSIC, UPV/EHU), 48940 Leioa, Spain

**Keywords:** atomic force microscopy, S100A9, biomolecular processes, oligomerization, protein fibrils, mechanical properties, fibril topography

## Abstract

Atomic force microscopy (AFM) imaging enables the visualization of protein molecules with high resolution, providing insights into their shape, size, and surface topography. Here, we use AFM to study the aggregation process of protein S100A9 in physiological conditions, in the presence of calcium at a molar ratio 4Ca^2+^:S100A9. We find that S100A9 readily assembles into a worm-like fibril, with a period dimension along the fibril axis of 11.5 nm. The fibril’s chain length extends up to 136 periods after an incubation time of 144 h. At room temperature, the fibril’s bending stiffness was found to be $2.95\times 10^{-28}$ Nm^2^, indicating that the fibrils are relatively flexible. Additionally, the values obtained for the Young’s modulus ($E_{x}=6.96\times 10^{5}$ Pa and $E_{y}=3.37\times 10^{5}$ Pa) are four orders of magnitude lower than those typically reported for canonical amyloid fibrils. Our findings suggest that, under the investigated conditions, a distinct aggregation mechanism may be in place in the presence of calcium. Therefore, the findings reported here could have implications for the field of biomedicine, particularly with regard to Alzheimer’s disease.

## 1. Introduction

Among the various techniques available for imaging biological systems, Atomic Force Microscopy (AFM) stands out due to its remarkable versatility, high resolution and precision. Since its invention in the mid-1980s [1], AFM imaging and spectroscopy have been used to study a variety of biological systems such as antigen/antibody binding [2], protein assemblies [3] and biomolecular interactions [4], lipid layers [5], DNA/small peptides [6], cells [7], bacteria [8,9], yeast [10,11] soft-matter systems [12] and also other systems such as nanocomposites [13] and lignocellulosic polymers [14]. AFM revolutionized the field of structural biology by providing high-resolution images of proteins in their native environments, without requiring extensive sample preparation [15]. When studying biomolecular systems, this feature makes AFM more attractive than other scanning probes microscopy (SPM) techniques, or electron microscopies (TEM) [16].

Protein aggregation is a complex biomolecular process that results from protein–protein association [17]. A possible outcome of protein aggregation is the formation of insoluble amyloid fibrils, which are often associated with the onset of the so-called conformational disorders. While Parkinson’s and Alzheimer’s diseases are, perhaps, the most well-known examples, more than 40 pathologies have been linked to amyloids [18]. According to recently published epidemiological data, there are 10 million new cases of neurological disorders worldwide [19], with 276 million of disability-adjusted life years (DALY) patients exhibiting cognitive decline [20]. Furthermore, it is anticipated that neurodegenerative illnesses will account for roughly 9.0 million deaths annually and that this number will nearly triple over the next three decades [21].

The outcome of protein self-association is not necessarily amyloid since it may also lead to amorphous aggregates lacking a well-defined structure. Intermediate species formed along the aggregation cascade include soluble and insoluble oligomers, protofibrils, and fibrils [22]. In the case of protein aggregation, AFM stands out as a particularly powerful tool, enabling the visualization and the characterization of protein aggregates with high resolution and detail. In this framework, it is relevant to use AFM with ultrashort cantilevers to gather AFM images in a few seconds (high-speed AFM) [23] or the molecular recognition imaging working at low-force regimes that makes it possible to localize the single protein:ligand interaction events [24].

Additionally, AFM provides accurate information about the size, shape, and morphology of oligomers, fibrils, and other aggregated species formed in real time [25].

Protein S100A9 is a member of the family of the so-called S100 proteins, which are calcium-binding proteins [26,27]. The latter are signaling molecules that play a plethora of intra- and extracellular functions, including signal transduction, cell differentiation, proteostasis, and inflammation, just to mention a few examples [28]. The S100A9 protein is abundant in the brain, and increased levels of S100A9 have been found in the brains of Alzheimer’s Disease (AD) patients [29], making it a robust biomarker for AD [30]. Some studies have shown that S100A9 can interact with other proteins and modulate their aggregation propensity (see, e.g., [31,32]). Moreover, it has been reported that S100A9 is able to rapidly form amyloid fibrils under physiological conditions in the absence of calcium [33]. Interestingly, recent results indicate that calcium concentration is an important modulator of the S100A9 fibrillation, being actually able to inhibit the formation of amyloid fibrils [34].

Here, we use AFM to study the dynamics, morphological and mechanical properties of oligomeric fibrils resulting from S100A9 aggregation in physiological conditions in the presence of calcium at a molar ratio of 4Ca^2+^:S100A9.

## 2. Materials and Methods

### 2.1. Sample Preparation

Human recombinant S100A9 protein was purified to homogeneity as previously described [35,36] and was a kind gift from Dr. Cláudio Gomes (BioISI, Faculdade de Ciências, Universidade de Lisboa), supplied in the native apo homodimeric state (100 μM) and as oligomers formed upon incubation of 100 μM S100A9 in 50 mM Tris-HCl, pH 7.4 with a 4-fold excess of CaCl_2_ at 37 °C under continuous agitation (250 rpm).

### 2.2. Mica Preparation and AFM Measurements

Approximately 10 μL of each S100A9 solution was deposited onto freshly cleaved mica (Agar Scientific, Stansted, UK). Before imaging, the samples were allowed to rest in air for about 20 min to ensure that both the fibrils and S100A9 proteins remained attached to the mica surface. Afterwards, they were rinsed with Milli-Q water to remove the buffer and salt components and dried in air at room temperature.

The topography/morphology of the samples was characterized by AFM using a Pico SPM LE (Molecular Imaging) system. The images were obtained in air at room temperature, first with HQ:NSC 35/Hard/Al BS (force constants 5.4 N/m, 8.9 N/m, and 16 N/m, tip radius < 20 nm) µmasch cantilevers in dynamic mode which reduces the damage to biological samples. Some selected samples were analyzed with HiRes-C14/Cr–Au (force constant 5 N/m, tip radius < 1 nm) probes which are optimized for imaging relatively soft samples and obtain a better phase contrast reducing surface deformations caused by the AFM tip. To confirm the height measurements of the structures detected under dry conditions, AFM images of the samples in an aqueous environment were also obtained.

### 2.3. Data Analysis

For each sample, the existing structures were systematically imaged, and the images analyzed. Images were processed using Gwyddion v2.56 [37].

#### 2.3.1. Height Measurements

Height measurements were made from scan sizes of 5 × 5 μm^2^, 2 × 2 μm^2^, or 1 × 1 μm^2^ by measuring height profiles normal to the fibril axis and using the full width at half maximum (FWHM) of the peak height of the profile. Heights for the different fibril species were obtained by averaging measurements from >20 individual fibrils, with errors being calculated as two standard deviations (2σ) from the mean.

#### 2.3.2. Mechanical Properties

In what follows, we model a protein fibril as a polymer chain in two dimensions. The bending stiffness, *B*, of a polymeric chain can be obtained from the persistence length, $L_{P}$, according to
(1)$$L_{P}=\frac{B}{k_{B}T},$$
where $k_{B}$ is the Boltzmann constant and *T* is the absolute temperature. It can be shown [38] that, in two dimensions, $L_{P}$ can be calculated by using the end-to-end distance, *R*, and the contour length, $L_{C}$, according to the equation
(2)$$<R^{2}>=4L_{P}L_{C}\left[1-\frac{2L_{P}}{L_{C}}\left(1-e^{\left(-L_{C}/2L_{P}\right)}\right)\right].$$
On the other hand, the Young’s modulus, *E*, can be obtained from the bending stiffness [39] by using
(3)B=EI,
where *I* is the area moment of inertia (or second moment of inertia) of the polymer chain. In the case of a filled ellipse with radius *a* along the *x*-axis and radius *b* along the *y*-axis with respect to an axis running through the center of the ellipse and parallel to the *x*-axis, the components of *I* are
(4)$$I_{xx}=\frac{\pi }{4}ab^{3},$$
and with respect to the perpendicular axis is
(5)$$I_{yy}=\frac{\pi }{4}a^{3}b.$$

The polymer’s contour length, $L_{C}$, and the end-to-end distance, *R*, were measured with Gwyddion [37].

To obtain the fibrils’ persistence length, $L_{p}$, a fit of the worm-like chain model (Equation 2) to the experimental data was performed with Mathematica [40].

## 3. Results and Discussion

### 3.1. The S100A9 Protein

S100 proteins adopt different quaternary structures, including homo- and heterodimers, and high order multimers [27]. The native structure of the S100A9 protein is an homodimer of two S100A9 monomers. Each monomer is composed of two EF-hand motifs (i.e., helix–loop–helix structures capable of binding calcium ions). The N-terminal motif comprises helices I and II and is connected to the C-terminal motif, which comprises helices III and IV, by a short flexible linker. The two EF-hand motifs provide two calcium (Ca^2+^) binding sites (Figure 1).

### 3.2. Formation of Fibrillar Oligomers

Mica surfaces with S100A9 were analyzed with AFM to investigate the dynamics of self-association starting from the apo S100A9 native dimers with a concentration of 100 μM (Figure 2A). The latter were incubated at 37 °C (pH 7.4) with an agitation speed of 250 revolutions per minute (rpm) for up to 48 h, giving rise to a heterogeneous equilibrium population of soluble oligomers differing in size and morphology. It is possible to observe relatively long, slender filaments with a worm-like shape (i.e., worm-like fibrils), as well as some closed-ring shapes and a few branched structures (Figure 2B).

It has been shown that Ca^2+^ ions play a crucial role in modulating the structure, function, regulation, signaling activity [35], and self-association [34] of S100A being able to suppress the formation of amyloid fibrils. Here, we focused on the self-association of S100A9 in the presence of calcium at a molar ratio of 4Ca^2+^:S100A9 to access its impact on the formation and morphology of fibrillar structures. We find that a similar equilibrium population is observed after 48 h both in the absence (Figure 2B) and in the presence of calcium (Figure 2C), indicating that, when present at the molar ratio of 4Ca^2+^:S100A9, calcium has no noticeable effect on the morphology of the oligomer’s population.

To gain further insight on the dynamics of self-association in the presence of calcium, we performed a time-resolved AFM analysis. In particular, AFM images were collected after 5 h, 15 h, 24 h, 48 h, 72 h, and 144 h (Figure 3). For each incubation time, at least 5 samples were prepared, and a minimum of 20 individual fibrils were analyzed.

It is possible to see that a significant number of the worm-like fibrils is already formed after 15 h, and that the former increases significantly afterwards. Indeed, due to the large quantity of material accumulated after 144 h of incubation time, obtaining high-quality images proved to be challenging. Consequently, we present an image of a sample fibril within a 1 × 1 μm^2^ area to provide an example of a typical observed structure.

### 3.3. Fibril Height

To measure the fibril’s height, we considered the images corresponding to the S100A9 + 4Ca^2+^ fibril populations formed after different incubation times. AFM imaging and AFM statistical analysis show that the fibril’s height is conserved over time. S100A9 self-associates into fibrils with median height in the AFM images of 2.9±0.8 nm after 24 h (dry) and 3.2±0.5 nm after 48 h (liquid) (Figure 4).

### 3.4. Period and Fibril Length

To obtain a more fine-grained picture of the worm-like fibrillar structures, we considered the images corresponding to 24 h because the number of species formed in the corresponding samples makes it possible to isolate and perform a detailed analysis of the structures therein. It is possible to see that the worm-like fibrillar structures appear to be tandem repeats of a geometrically regular unit (Figure 5) with dimensions 12 ± 2 nm (width) and 11.5 ± 1 nm (width along the other direction), which will be taken as the fibril period. Having identified the fibril period, we take the fibril length as the number of periods. We observe a moderate linear correlation between fibril length and incubation time (Figure 6) starting from 6 periods (5 h) to 136 periods (144 h).

### 3.5. Mechanical Properties and Biophysical Characterization of Fibrillar Oligomers

Assuming that the fibrils interact weakly with the mica surface and relax to a two-dimensional equilibrium conformation, the persistence length can be obtained from the shape of the fibril, as given by Equation (Equation 2). To measure $L_{P}$, we considered a sample composed of 354 fibrils obtained after 48 h. The contour length and end-to-end distance of each fibril were measured with Gwyddion. Maximum height measurements were used to characterize the fibril’s cross-sections. The latter were used to evaluate the moments of inertia as given by Equations (Equation 4) and (Equation 5). By performing a statistical analysis of AFM images, we were able to precisely determine the cross-section of S100A9 fibrils.

Figure 7 reports the dependence of $<R^{2}>$ on $L_{C}$.

The persistence length, $L_{P}$, characterizes the flexibility and stiffness of a polymer chain. It quantifies how effectively the polymer maintains its initial direction over a certain distance upon being perturbed by thermal energy. A longer persistence length indicates a stiffer polymer chain, i.e., one that tends to maintain its shape and orientation over greater distances, while a shorter persistence length implies a more flexible and easily deformable polymer chain. In the S100A9 fibrils, the measured persistent length is $L_{P}=71.2$ nm (σ=1.6 nm), showing that they are relatively flexible. This value of $L_{P}$ is of the same order of magnitude but 23% smaller as that reported in [43] for worm-like β-lactoglobulin amyloid fibrils, and 1.7 orders of magnitude smaller that the persistence length of straight amyloid fibrils formed by the same protein.

The bending stiffness describes a material’s resistance to bending deformation when subjected to an external force. It represents the ability of a structural element to resist bending and maintain its shape under bending loads. At room temperature, the bending stiffness was found to be $2.95\times 10^{-28}$ Nm^2^ (Figure 7). This rather low value supports the view that the S100A9 fibrils are relatively flexible and can easily undergo bending deformations.

The Young’s modulus quantifies the stiffness of a material and its resistance to deformation when subjected to tensile or compressive forces within the elastic range. It is a scalar quantity that represents the ratio of stress to strain within the material. We found that, along the *x* direction, the Young’s modulus is $E_{x}=6.96\times 10^{5}$ Pa, being $E_{y}=3.37\times 10^{5}$ Pa along the *y* direction (with an uncertainty lower than $1\times 10^{5}$ Pa). These values for the Young’s modulus are orders of magnitude lower than those reported in the literature for amyloid fibers (3–20 GPa) [25,44,45], indicating an atypical nature of S100A9 fibrils formed in the presence of calcium.

## 4. Conclusions

Conventional AFM imaging makes it possible to identify and characterize distinct species formed at particular stages of protein aggregation [46,47]. It was reported that, in the absence of calcium, S100A9 self-associates into amyloid fibrils following a two-step nucleation-autocatalytic growth model [33]. Here, we used AFM imaging to explore the aggregation pathway of the S100A9 protein in the presence of calcium in physiological conditions. We found that, in these conditions, S100A9 readily forms worm-like fibrils. Furthermore, the average fibril’s length, measured as the number of periods, exhibits a moderately linear dependence on the incubation time. By calculating selected mechanical properties, we found that the fibrils are rather flexible, being able to undergo bending deformations. The fibril’s mechanical properties, in particular the Young’s modulus, are quantitatively distinct from those of typical amyloid fibrils, suggesting that calcium may modulate S100A9 self-association and assembly. Previous reports [34] showed that the presence of calcium at high concentration (in the mM range and, therefore, well above the μM physiological threshold [48]) inhibits the formation of S100A9 amyloid fibrils. Overall, these results suggest that, in the presence of calcium, S100A9 worm-like fibrils may follow a different aggregation pathway, presumably because they possess unique structural characteristics compared to amyloid fibrils. Further research into the specific structural features (e.g., to obtain a fine-grained description of the fibril’s period) and functional implications of these worm-like fibrils could shed light on their biological significance and potential applications in the field of biomedicine, in particular in the context of Alzheimer’s disease.

## Figures and Tables

**Figure 1 biomolecules-14-01091-f001:**
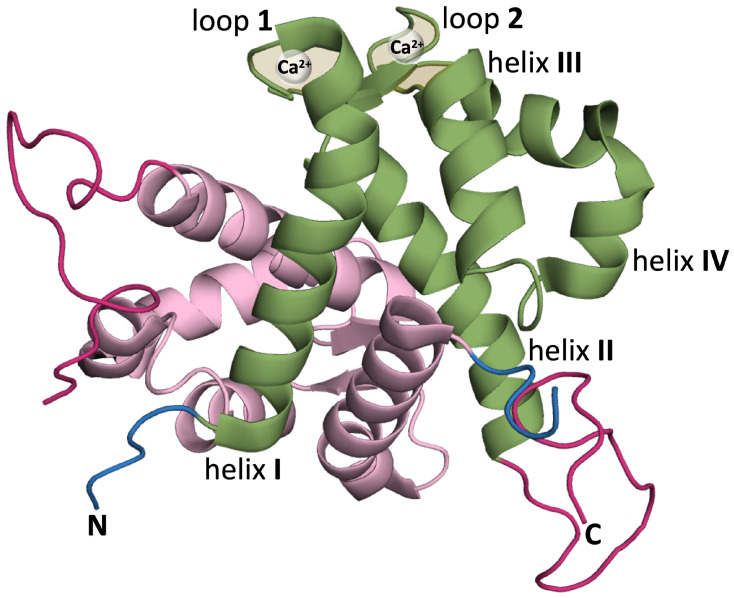
Ribbon representation of homodimeric S100A9 protein (PDB:ID 5I8N) [41] using PyMol [42] (version 2.4.0) with one of the monomers colored in green and the other in pink. The monomer colored in green highlights helix I (Q7-S23), helix II (Q34-D44), calcium binding loop 1 (V24-N33), helix III (E56-L66) (helix E of canonical EF hand), calcium binding loop 2 (D67-S75), and helix IV (F76-M94) (helix F of canonical EF hand) [28]. In each monomer, the N-terminal is shown in blue and the (unstructured) C-terminus in dark pink.

**Figure 2 biomolecules-14-01091-f002:**
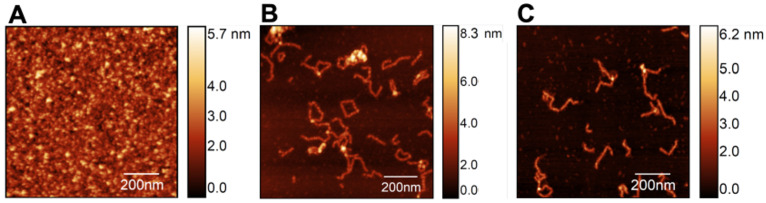
AFM imaging of S100A9 fibrillar structures formed at 37 °C (pH 7.4) starting from native dimers (**A**) incubated in the absence (**B**) and in the presence (**C**) of calcium at a molar ratio of 4Ca^2+^:S100A9. Scan sizes are 1 × 1 μm^2^.

**Figure 3 biomolecules-14-01091-f003:**
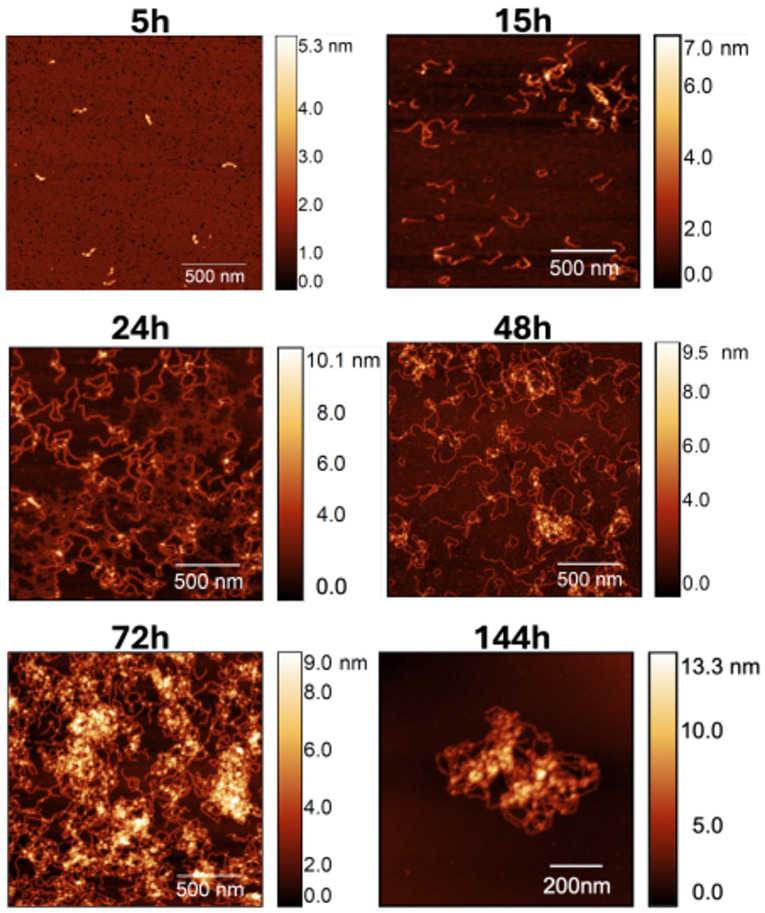
AFM images of S100A9 fibril population observed for increasing incubation time in the presence of calcium at a molar ratio of 4Ca^2+^:S100A9. Scan sizes are 2 × 2 μm^2^ for 5, 15, 24, 48, and 72 h, and 1 × 1 μm^2^ in the 144 h image.

**Figure 4 biomolecules-14-01091-f004:**
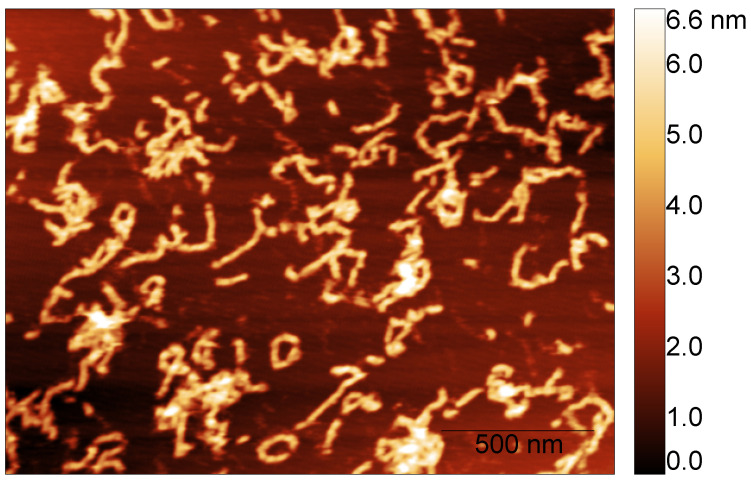
Example of S100A9 fibril population imaged in liquid after 48 h of incubation time in the presence of calcium at a molar ratio of 4Ca^2+^:S100A9.

**Figure 5 biomolecules-14-01091-f005:**
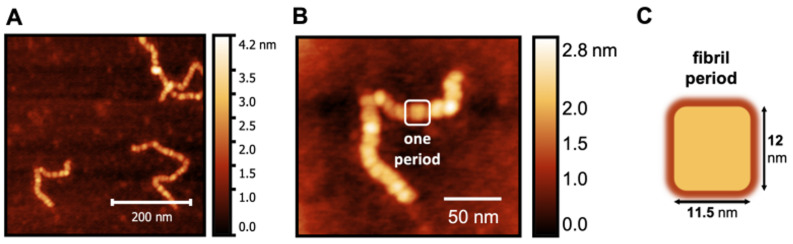
AFM image of a magnified S100A9 fibril (**A**,**B**) formed after 24 h showing that it consists of a tandem repeat of geometrically regular units that we identify as the fibril period (**C**).

**Figure 6 biomolecules-14-01091-f006:**
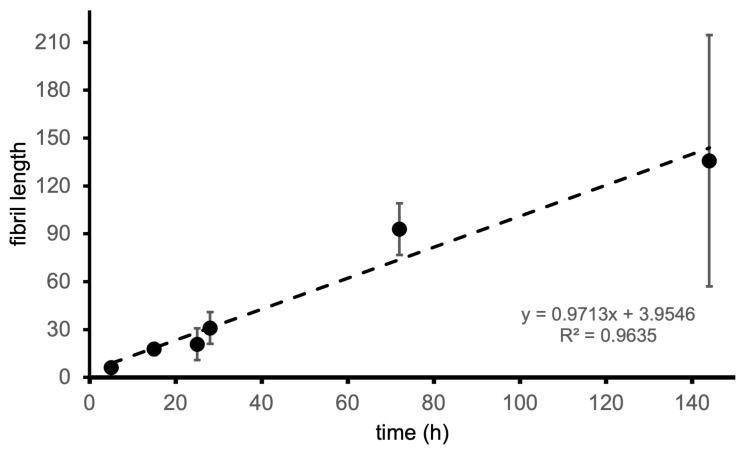
Fibril length (measured as the number of periods) as a function of incubation time.

**Figure 7 biomolecules-14-01091-f007:**
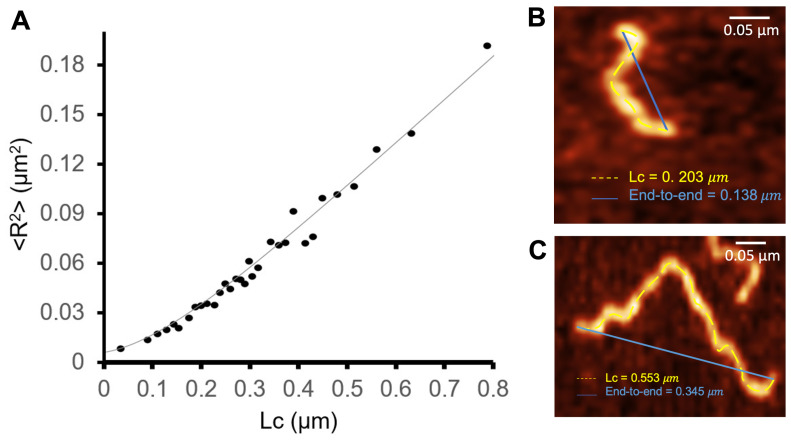
Dependence of the mean square value of the end-to-end distance as a function of the contour length. The fit corresponds to the worm-like chain model in two dimensions, $R^{2}\left(fit\right)=0.9935$; $L_{p}=71.2$ nm (σ = 1.6 nm); $S_{b}=2.95\times 10^{-28}$ Nm^2^; $L_{p}$ was used as a fitting parameter (**A**). The end-to-end distance and contour length (**B**,**C**) were measured with Gwyddion in images obtained with 48 h of incubation time.

## Data Availability

Data are contained within the article.

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
