# Peer review of "Morphological and Biophysical Study of S100A9 Protein Fibrils by Atomic Force Microscopy Imaging and Nanomechanical Analysis"

_biomolecules, 2024, doi:10.3390/biom14091091_

Round 1

Reviewer 1 Report

Comments and Suggestions for Authors

The manuscript entitled “Morphological and biophysical analysis of S100A9 protein fibrils by Atomic Force Microscopy” by A. P. Carapeto et al. reports on the use of atomic force microscopy (AFM) in studying the dynamics, morphology, and mechanical properties of S100A9 protein fibrils. It focuses on the properties of fibrils formed in the presence of moderate calcium concentration under physiological conditions. AFM image analysis allows the quantification of fibril morphology (length, height, period) and mechanical properties (persistence length, Young's modulus), providing valuable insight into the aggregation pathway of S100A9 protein at given conditions. Overall, the manuscript presents valid results suggesting a unique aggregation mechanism of S100A9 in the presence of moderate calcium concentrations. Further research into the specific structural features and functional implications of these worm-like fibrils is important for determining their biological significance.

 I recommend acceptance after minor revisions that clarify the following methodological details:

1.       In section 3.2, the authors stated that the heterogenous equilibrium population of soluble oligomers was observed. Since the amyloid aggregation kinetic is shown only by AFM, information about how many independent samples of fibrils were prepared (to confirm the reproducibility) is important to strengthen the statement. Similarly, how many individual fibrils were used for analysis should be added. (The information about the number of fibrils is included only for persistence length calculations).

 2.       In section 2.2, the authors stated that they used 2 types of probes, each with a different force constant and tip radius, with the vague statement that “some samples” were analyzed with HiRes probes. This should be clarified in detail.

 This also implies the important question of why the authors did not include the correction of topographic parameters in dependence on tip size. Even though the authors used images obtained in dry and liquid conditions to partially overcome this problem, the potential influence of tip size on topographic parameters should be discussed.

Author Response

REVIEWER 1

1.In section 3.2, the authors stated that the heterogenous equilibrium population of soluble oligomers was observed. Since the amyloid aggregation kinetic is shown only by AFM, information about how many independent samples of fibrils were prepared (to confirm the reproducibility) is important to strengthen the statement. Similarly, how many individual fibrils were used for analysis should be added. (The information about the number of fibrils is included only for persistence length calculations).

Response: The requested information details were added in the lines 138-139: “For each incubation time at least 5 samples were prepared and at least 20 individual fibrils were analyzed.”

2.In section 2.2, the authors stated that they used 2 types of probes, each with a different force constant and tip radius, with the vague statement that “some samples” were analyzed with HiRes probes. This should be clarified in detail.

This also implies the important question of why the authors did not include the correction of topographic parameters in dependence on tip size. Even though the authors used images obtained in dry and liquid conditions to partially overcome this problem, the potential influence of tip size on topographic parameters should be discussed.

Response: The reviewer is right. The samples analyzed by HiRes probes were a selection of the more representative for each condition. The AFM tip convolution effects is inherent to this type of measurements. For this reason, ultrasharp AFM tips with nominal radius of 1 nm were used in order to minimize this detrimental effect. The same tip was used for all the AFM imaging experiments in order to get comparable the gathered data of the visualized S100A9 fibrils among the different incubation times.

Reviewer 2 Report

Comments and Suggestions for Authors

The manuscript titled “Morphological and biophysical analysis of S100A9 protein fibrils by atomic force microscopy” by Carapeto, A.P.; et al. is a scientific work where the authors assessed the changes in terms of S100A9 fibril dimensions and rigidity at the single molecule level with ultrasensitive atomic force microscopy (AFM) measurements. The novelty of this work is high and the outcomes found by the authors could have a strong significance to better understand the underpinning molecular mechanisms associated to protein fibrillization processes which could lead to neurodegenerative malignancies. This is a topic of growing interest. However, it exists some points that need to be addressed (please, see them below detailed point-by-point) to improve the scientific quality of the submitted manuscript paper before this article will be consider for its publication in Biomolecules.

1) TITLE. There exists many AFM operational modes. It may be desirable to indicate the type of analysis carried out by the authors in this research. For this reason, it may be more appropriate to modify the current paper title by “Morphological and biophysical analysis of S100A9 protein fibrils by atomic force microscopy imaging and nanomechanics”

2) Abstract. “AFM imaging enables (…)” (line 1). The full name should be stated (atomic force microscopy). Then, the abbreviation should be placed between brackets. This comment should be taken into consideration for the rest of the main manuscript body text.

3) The authors should consider to add the terms “fibril topology” and “mechanical properties” in the keyword list.

4) “A possible outcome of protein aggregation (…) While Parkinson’s and Alzheimer’s diseases are, perhaps, the most well-know examples, more than 40 pathologies have been linked to amylodis” (lines 27-30). Firstly, the term “well-know” should be exchanged by “well-known”. Even, if the manuscript is generally well-written, it is necessary to revisit it in order to polish many English grammar expressions. Then, could the authors provide quantitative insights about the worldwide global burdens of amyloidogenic diseases? This will significantly aid the potential readers to better understand the significance of this devoted research.

5) “In the case of protein aggregation, AFM stands out as a particularly powerful tool (…) aggregated species formed in real time” (lines 35-38). Here, it may be desirable to expand the discussion concerning the AFM capabilities and its working principles. In this framework, it is relevant the use of AFM with ultrashort cantilevers to gather AFM images in few seconds (high-speed AFM) [1] or the molecular recognition imaging working at low force regimes that enables to localize the single protein:ligand interaction events [2].

[1] Ando, T. Functional Implications of Dynamic Structures of Intrinsically Disordered Proteins Revealed by High-Speed AFM Imaging. Biomolecules 2022, 12, 1876. https://doi.org/10.3390/biom12121876

[2] Marcuello, C.; de Miguel, R.; Lostao, A. Molecular Recognition of Proteins through Quantitative Force Maps at Single Molecule Level. Biomolecules 2022, 12, 594. https://doi.org/10.3390/biom12040594 (This work is even from one of the co-authors of this manuscript).

6) “4-fold excess of CaCl2” (line 60). Please, the chemical formula stoichiometry needs to be stated in subscript. This comment should be taken into  account for the rest of the main manuscript body text.

7) Figure 2 (line 120). The significant figures displayed in the vertical bars should be homogenized. Same comment for the Fig. 3 (line 121) and the Fig. 5, panels a-c (line 153).

8) “3.3. Fibril height and 3.4. Period and fibril length” (lines 112-152). Here, it may be desirable to add a Table to summarize all the data concerning to the fibril height and length dimensions achieved at the different incubation times tested in this research.

9) “3.5. Mechanical properties and biphysical characterization of fibrillar oligomers” (lines 153-186). Did the authors conduct nanoindentation measurements? In case affirmative, the stifness maps (and the respective topography images) should be furnished in this section.

10) “Conclusions” (lines 187-207). This section perfectly remarks the most relevant outcomes found by the authors in this work and the promising future perspectives. It should be desirable to add a brief statement to discuss about the potential future action lines to pursue the topic covered in this research

Comments on the Quality of English Language

The manuscript is generally well-written albeit it may be desirable if the authors could extensively revisit it in order to polish those final details susceptible to be improved.

Author Response

REVIEWER 2

1)TITLE. There exists many AFM operational modes. It may be desirable to indicate the type of analysis carried out by the authors in this research. For this reason, it may be more appropriate to modify the current paper title by “Morphological and biophysical analysis of S100A9 protein fibrils by atomic force microscopy imaging and nanomechanics”

Response: We agree with this comment provided by the reviewer and changed the title accordingly.

2) Abstract. “AFM imaging enables (…)” (line 1). The full name should be stated (atomic force microscopy). Then, the abbreviation should be placed between brackets. This comment should be taken into consideration for the rest of the main manuscript body text.

Response: Thank you for pointing this out. It has been corrected.

3) The authors should consider adding the terms “fibril topology” and “mechanical properties” in the keyword list.

Response: We thank the reviewer for this suggestion. We expanded the keyword list, but instead of fibril topology we have added fibril topography, since no topological analysis was performed in this study.

4) “A possible outcome of protein aggregation (…) While Parkinson’s and Alzheimer’s diseases are, perhaps, the most well-known examples, more than 40 pathologies have been linked to amyloids” (lines 27-30). Firstly, the term “well-know” should be exchanged by “well-known”. Even, if the manuscript is generally well-written, it is necessary to revisit it in order to polish many English grammar expressions. Then, could the authors provide quantitative insights about the worldwide global burdens of amyloidogenic diseases? This will significantly aid the potential readers to better understand the significance of this devoted research.

Response: Thank you for pointing out the typo: The word “well-know” was modified by “well-known”. To broaden the scholarship of the manuscript we added the following sentence to the introductory section: “According to recently published epidemiological data, there are 10 million new cases of neurological disorders worldwide [19], with 276 million of disability-adjust life-years (DALY) patients exhibiting cognitive decline [20]. Furthermore, it is anticipated that neurodegenerative illnesses will account for roughly 9.0 million deaths annually and that this number will nearly triple over the next three decades [21].”

5) “In the case of protein aggregation, AFM stands out as a particularly powerful tool (…) aggregated species formed in real time” (lines 35-38). Here, it may be desirable to expand the discussion concerning the AFM capabilities and its working principles. In this framework, it is relevant the use of AFM with ultrashort cantilevers to gather AFM images in few seconds (high-speed AFM) [1] or the molecular recognition imaging working at low force regimes that enables to localize the single protein:ligand interaction events [2].

[1] Ando, T. Functional Implications of Dynamic Structures of Intrinsically Disordered Proteins Revealed by High-Speed AFM Imaging. Biomolecules 2022, 12, 1876. https://doi.org/10.3390/biom12121876

[2] Marcuello, C.; de Miguel, R.; Lostao, A. Molecular Recognition of Proteins through Quantitative Force Maps at Single Molecule Level. Biomolecules 2022, 12, 594. https://doi.org/10.3390/biom12040594 (This work is even from one of the co-authors of this manuscript).

Response: We thank the reviewer for his comment and suggestion to provide more detail on the use of AFM to study protein aggregation. The references indicated by the reviewer were included, and the text now reads:

The outcome of protein self-association is not necessarily amyloid since it may also lead to amorphous aggregates lacking a well-defined structure. Intermediate species formed along the aggregation cascade include soluble and insoluble oligomers, protofibrils and fibrils \cite{dobson2003protein}. In the case of protein aggregation, AFM stands out as a particularly powerful tool, enabling the visualization and the characterization of protein aggregates with high resolution and detail. In this framework, it is relevant the use of AFM with ultrashort cantilevers to gather AFM images in few seconds (high-speed AFM) [23] or the molecular recognition imaging working at low force regimes that enables to localize the single protein: ligand interaction events [24].

6) “4-fold excess of CaCl2” (line 60). Please, the chemical formula stoichiometry needs to be stated in subscript. This comment should be considered for the rest of the main manuscript body text.

Response: Thank you for pointing this out. All the chemical formulas were checked, and the stoichiometry coefficients are displayed in lowercase.

7) Figure 2 (line 120). The significant figures displayed in the vertical bars should be homogenized. Same comment for the Fig. 3 (line 121) and the Fig. 5, panels a-c (line 153).

Response: We appreciate this comment from the Reviewer. The vertical scale bars of the Figures 2, 3 and 5 were modified accordingly.

8) “3.3. Fibril height and 3.4. Period and fibril length” (lines 112-152). Here, it may be desirable to add a Table to summarize all the data concerning to the fibril height and length dimensions achieved at the different incubation times tested in this research.

Response: The fibril height was conserved in the samples corresponding to the different incubations times and this is why we only provided the median height for the population corresponding at 24h (2.9 ± 0.8 nm (dry)) and at 48h (3.2± 0.5 nm (liquid)). To convey a clearer message, we re-wrote the first sentence of section 3 that now reads: To measure the fibril’s height we considered the images corresponding to the S100A9 + 4Ca2+ fibril equilibrium populations formed after different incubation times. AFM imaging and AFM statistical analysis shows that the median fibril’s height is conserved along time. S100A9 self-associates into fibrils with median height in the AFM images of  2.9 ± 0.8 nm after 24h (dry), and 3.2± 0.5 nm after 48h (liquid)

9) “3.5. Mechanical properties and biophysical characterization of fibrillar oligomers” (lines 153-186). Did the authors conduct nanoindentation measurements? In case affirmative, the stifness maps (and the respective topography images) should be furnished in this section.

Response: In this work, the Young’s modulus (or elastic modulus) was extracted from the fibrilar bending stiffness that directly relies on the fibril contour length and the end-to-end distance values as indicated in the equations 1-3.

10)“Conclusions” (lines 187-207). This section perfectly remarks the most relevant outcomes found by the authors in this work and the promising future perspectives. It should be desirable to add a brief statement to discuss about the potential future action lines to pursue the topic covered in this research.

Response: On our view the next step is to investigate in more detail the structural features of the reported wormlike fibrils. In particular, to get a fine-grained description of the fibril’s period. Accordingly, the conclusions section ends with the following statement: Further research into the specific structural features (e.g., to obtain a fine-grained description of the fibril’s period) and functional implications of these worm-like fibrils could shed light on their biological significance and potential applications in the field of biomedicine, in particular, in the context of Alzheimer’s disease.

Round 2

Reviewer 2 Report

Comments and Suggestions for Authors

The authors did a great deal of effort to cover all the suggestions raised by the Reviewers. For this reason, the scientific manuscript quality was greatly improved. Based on the novelty and significance of the most relevant outcomes found in this research, I warmly endorse this work for further publication in Biomolecules.